# Defining the role of the polyasparagine repeat domain of the *S. cerevisiae* transcription factor Azf1p

**Taylor Stewart[1], Benjamin E. Wolfe[1], Stephen M. Fuchs**[1,2,3]*

**1** Department of Biology, Tufts University, Medford, MA, United States of America, **2** Institute for Protein Innovation, Boston, MA, United States of America, **3** Program in Cellular and Molecular Medicine, Boston Children's Hospital, Boston, MA, United States of America

* stephen.fuchs@proteininnovation.org

## Abstract

Across eukaryotes, homopolymeric repeats of amino acids are enriched in regulatory proteins such as transcription factors and chromatin remodelers. These domains play important roles in signaling, binding, prion formation, and functional phase separation. Azf1p is a prion-forming yeast transcription factor that contains two homorepeat domains, a polyglutamine and a polyasparagine domain. In this work, we report a new phenotype for Azf1p and identify a large set of genes that are regulated by Azf1p during growth in glucose. We show that the polyasparagine (polyN) domain plays a subtle role in transcription but is dispensable for Azf1p localization and prion formation. Genes upregulated upon deletion of the polyN domain are enriched in functions related to carbon metabolism and storage. This domain may therefore be a useful target for engineering yeast strains for fermentation applications and small molecule production. We also report that both the polyasparagine and polyglutamine domains vary in length across strains of *S. cerevisiae* and propose a model for how this variation may impact protein function.

## Introduction

Repeat domains are prevalent in the eukaryotic proteome and are found in 13% of proteins in yeast, 17% in humans, and 21% in *Drosophila*. Repeats are commonly found within intrinsically disordered regions (IDRs), which are protein domains that do not have a single native fold corresponding to a lowest energy state. Instead, these domains dynamically sample a number of conformations with similar energies [1]. Despite being historically dismissed as unimportant due to their lack of three-dimensional structure, repetitive protein domains are essential for binding, signaling, and functional phase separation. Across eukaryotes, repeats are enriched in cell-surface and regulatory proteins, such as transcription factors (TFs) and chromatin remodelers [2]. Repeats enriched in a single amino acid are deemed homorepeats. Polyglutamine (polyQ) is the most common type of homorepeat in eukaryotic proteins, followed by polyasparagine (polyN). While polyQ domains are found across eukaryotes, polyN domains are rare in vertebrates and found almost exclusively in more primitive invertebrates

**Data Availability Statement:** All relevant data are within the paper and its Supporting Information files. Strains and plasmids are available upon request.

**Funding:** This work was supported by the Army Research Office [W911NF-16-1-0175, W911NF-19-1- 0299, and W911NF-20-1-0083 to S.M.F.]

**Competing interests:** The authors have declared that no competing interests exist.

[3]. Synthetic polyN peptides have been shown to mediate aggregation [4,5], but the role of natural polyN domains in protein function has only been explored in a few cases. For example, the polyN domain of Pdr1p has been shown to be important for transcriptional activation [6]. Whether or not this function expands to other polyN-containing TFs, such as *CCR4, MIT1, ASG1, HAP4, ABF1, MOT3, AZF1,* and *RLM1,* has not been investigated.

Several polyQ and polyN-containing TFs in yeast (*HAP4, AZF1, MOT3, RLM1, CYC8, MGA1,* and *SWI1*) have been demonstrated to form prions. Prions are proteins that have multiple conformations, at least one of which is self-templating and epigenetically heritable. Prion-forming domains (PrDs) are characterized by low-complexity amino acid sequences that are often rich in N and Q, insensitive to scrambling, and intrinsically disordered in their native state [7]. Many prion-forming proteins contain repeats that are important to the function of both the prion and the native protein. Although there is only one bona fide prion in mammals, PrP<sup>Sc</sup>, screens to identify yeast prions have revealed that they are common in both lab and wild strains of yeast [8–11]. While the mammalian PrP<sup>Sc</sup> and prion-like proteins are well documented to cause both infectious diseases and age-related neurodegeneration in mammals [12–18], there is mounting evidence that yeast prions act as adaptive strategies, especially during environmental stress. Yeast prions are well-suited to mediate adaptation due to their enrichment in TFs and other regulatory proteins [10,11,19–29]. A screen conducted by Chakrabortee and colleagues identified nearly 50 novel prion-like proteins in yeast that conferred epigenetically heritable phenotypes after transient overexpression. In many cases, the prion state improved yeast growth rate in different environmental stresses, including osmotic and acidic stress. Like previously identified prions, many of the proteins identified in this screen are TFs and RNA-binding proteins [10].

Prion-forming TFs have the capacity to impact yeast growth through multiple mechanisms. These proteins can alter gene expression in both their native and prion states, leading to changes cellular functions such as metabolism, surface display, and stress tolerance. These proteins are therefore important targets for yeast metabolic engineering. Yeast are used in a wide range of applications in research, food and drink production, and small molecule production. Because homorepeat domains are at the intersection of TFs and prions, these domains are particularly interesting targets for engineering. Target genes of Q-rich TFs in yeast have significantly higher levels of expression divergence, expression variability, mutational variance, and expression noise when compared to targets of non-repeat-containing TFs [30]. These results highlight the importance of repeat domains in modulating gene expression.

The yeast TF Azf1p was recently shown to form a prion [10]. Azf1p has a largely nonoverlapping set of target genes during growth in fermentable and non-fermentable carbon sources, and the null mutant has been shown to have a severe growth defect in glycerol. *AZF1* is also important for cell wall integrity [31]. The prion conformer, designated [*AZF1*<sup>+</sup>], confers resistance to the drug radicicol in a gain-of-function manner but decreases the expression of Azf1p's target genes [10]. Azf1p contains a polyN and polyQ domain, and the frequency of homorepeats in yeast TFs and PrDs predicts a role for these domains in transcription as well as prion formation. In this work, we investigated the function of the polyN domain of Azf1p and find that this domain plays a subtle role in transcription but is not required for Azf1p localization or prion formation.

## Methods

### Yeast strains and plasmids

Plasmids used in this work are summarized in S1 Table. The ΔN mutant plasmid was constructed by QuikChange site-directed mutagenesis using pBY011-AZF1 (Harvard PlasmID) as

a template with the primers described in S2 Table. Yeast were grown on synthetic complete (SC) dropout medium, YPD (yeast extract, peptone, 2% dextrose), or YPG (yeast extract, peptone, 4% glycerol) at 30˚C. Plasmids were freshly transformed into yeast and maintained on SC media lacking uracil (SC-Ura). Strains used in this study are summarized in S3 Table and were derived from BY4741 (*MATa his3Δ1 leu2Δ0 met15Δ0 ura3Δ0*). The WT strain is from the yeast GFP collection in which the *AZF1* open reading frame is C-terminally tagged with GFP [32,33]. *AZF1ΔN* was constructed in this background using CRISPR [34]. A guide RNA sequence designed to target the polyQ domain of *AZF1* was cloned into the Cas9-expressing plasmid pML104. A ~1 kb fragment of *AZF1* containing the polyN deletion was amplified by PCR with 8 identical reactions using the AZF1 F/R primers (S2 Table) and pBY011-ΔN as a template. Ethanol-precipitated PCR products were co-transformed with pML104-polyQ, and cells were plated on SC-Ura. After 5 days, individual transformants were selected and grown overnight in 1 mL YPD in a 96 deep-well plate to allow for loss of the Cas9 plasmid. Overnight cultures were diluted ten-fold four times and spotted on FOA. Fast-growing FOA-resistant colonies were selected and characterized by colony PCR with the AZF1 F/R primers to identify yeast with the polyN deletion. This strain is referred to as ΔN in the text. *azf1-Δ1::URA3* was constructed in BY4741 using heterologous gene replacement with the *URA3* gene as a selectable marker. This strain was intended to be a full deletion of the *AZF1* open reading frame; however, *URA3* is instead integrated between *AZF1* and its promoter, abolishing *AZF1* expression (S4 Table). In the text, this strain is referred to as *azf1Δ* because it mimics the phenotypes associated with deletion of the *AZF1* open reading frame.

## Spotting assays

For phenotypic growth assays, yeast were grown overnight in YPD. Saturated overnight cultures were used to start fresh cultures in the same medium at an $OD_{600}$ of 0.2. The cells were allowed to double at least two times before approximately $1.0 \times 10^7$ cells were harvested and resuspended in sterile water in a 96-well plate. Cells were serially diluted five-fold five times and then spotted onto YPD or YPG plates using a 48-pin replicating tool. Plates were incubated at 30˚C and imaged after three (YPD) to seven (YPG) days.

## Growth curve assays

Each pBY011-AZF1 plasmid (WT and ΔN) was transformed into the matched background and maintained on SC-Ura (i.e. the ΔN plasmid was transformed into *AZF1ΔN*). Three biological replicates of each strain were grown overnight in SCRaff-Ura (2% raffinose). 2 μL of each culture were then transferred to 200 μL of four types of fresh media in a 96-well plate: SCGal-Ura (2% galactose), SCGal-Ura+Radicicol (75 μM), SC-URA, and SC-Ura+Radicicol. Growth was measured for 24 hours by $OD_{600}$ in a Molecular Devices SpectraMax M5 plate reader at 30˚C with shaking prior to each reading. Readings were taken every 30 minutes. 10 μL of each SCGal-Ura and SC-Ura culture were then transferred to 1 mL fresh SC-URA and allowed to grow to saturation overnight. The plate reader assay was then repeated in SC-URA media with and without radicicol. We compared the growth of cells whose ancestors had experienced *AZF1* overexpression to those that did not in order to determine the role of the polyN domain in [*AZF1*⁺] prion formation and activity.

A modified version of this plate reader assay was also performed with the WT, ΔN, and *azf1Δ* strains without the galactose-inducible plasmids. Three biological replicates were grown overnight in YPD. 2 μL of each saturated culture were then transferred to 200 μL of SC media, and growth was measured for 24 hours in the plate reader.

## RNA-seq

Three biological replicates were grown overnight in either YPD or YPG. Saturated overnight cultures were diluted to $OD_{600}$ 0.2. Cells were grown to $OD_{600}$ 0.6–1.0 and collected by centrifugation. Total RNA was isolated using the GE Healthcare RNAspin Mini RNA Isolation Kit per the manufacturer's guidelines and was quantified by NanoDrop (Thermo Fisher Scientific). mRNA was isolated using a NEBNext poly(A) mRNA magnetic isolation module (New England Biolabs). mRNA was used to generate RNA-seq libraries using a NEBNext Ultra II RNA library prep kit for Illumina (New England Biolabs) following the manufacturer's recommended protocol with the following parameters: 15 minutes at 42˚C for the RNA fragmentation time, a 1:25 dilution of the adapter, no size selection, and 15 cycles of denaturation/annealing/extension during the PCR enrichment step. After quantifying the library concentration with a Qubit HS DNA assay and determining the average peak size using a Fragment Analyzer, equimolar concentrations of each library were pooled together and sequenced on an Illumina NextSeq 550 at the Tufts University Genomics Core using a high output run with 150 cycles.

Data analysis was performed using Geneious Prime v. 2020.2.4 (Biomatters Ltd.). Sequencing reads were aligned to the s288c genome using the Geneious RNA Mapper with Medium-Low Sensitivity/Fast settings. Multiple best matches were mapped randomly. Expression levels were then calculated, and ambiguously mapped reads were counted as multiple full matches. To identify differentially expressed genes, expression levels were compared between strains in each carbon source using DEseq2 with a parametric fit type. Expression levels were also compared between carbon sources for each strain. Genes that are statistically significantly differentially expressed ($p<0.001$) in *azf1Δ* compared to WT were selected for GO Enrichment Analysis. The genes in this subset all have log2 ratios with a magnitude of 0.30 or greater. Because very few genes are statistically significantly differentially expressed in ΔN, genes with log2 fold ratios of ±0.30 or more were selected for analysis. GO Enrichment Analysis was performed using PANTHER v. 14 [35–37]. The PANTHER Overrepresentation Test (Released 20200728) employs a Fisher's exact test and reports significant enrichment in biological processes ($p<0.05$).

## Target gene expression measurements by RT-qPCR

RNA was isolated as described above, and cDNA synthesis was performed using the Super-Script™ First-Strand Synthesis System (Invitrogen) following the manufacturer's guidelines with an oligo-dT(20) primer and 300 ng of RNA per sample. cDNA was diluted 1:5 and stored at -20˚C. Each 20 µL reaction was prepared using the Brilliant II SYBR Green QPCR system (Agilent) following the manufacturer's guidelines with the primers listed in S2 Table. Samples were contained in MicroAmp 96-Well Reaction Plates (Applied Biosystems) sealed with MicroAmp Optical Adhesive Film (Applied Biosystems). Amplification was performed using an Applied Biosystems 7300 Real-Time PCR System with the following cycling conditions: 5 minutes at 95˚C and 40 cycles of 10 seconds at 95˚C followed by 30 seconds at 55˚C. The reference gene ACT1 was used for relative quantification. Gene expression changes were calculated using the $2^{-\Delta\Delta Ct}$ method. Two independent experiments were performed with three technical replicates for each biological replicate.

## Microscopy

Cells in which *AZF1* is tagged with GFP were imaged on a Leica Thunder Imager in the log and stationary phases of growth with either glucose or glycerol as the carbon source. WT and ΔN cells were grown overnight in YPD or YPG. 100 µL of saturated YPD overnight cultures

were spun down and imaged to study cells in stationary phase with glucose as the carbon source. Saturated overnight YPD cultures were also diluted to an $OD_{600}$ of 0.2 and grown to mid-log phase. 1.0 mL of these cultures was spun down and imaged to study cells growing in log phase with glucose as the carbon source. YPG cultures reached mid-log phase after about 24 hours. 1.0 mL of these cultures were spun down and imaged to study cells growing in log phase with glycerol as the carbon source. YPG cultures were also allowed to grow for 48 hours, at which time 100 µL were spun down and imaged to study cells growing in stationary phase with glycerol as the carbon source. Each condition was imaged two independent times, and at least 50 cells were imaged for each strain in each condition. Images were processed using Leica LAS X Instant Computational Clearing.

### Evaluation of AZF1 repeat variation

Variation within *AZF1* was measured using the data available from 93 recently sequenced *S. cerevisiae* genomes [38] as described in Babokhov *et al.* 2018 [47].

## Results

Azf1p is a prion-forming yeast TF that has been shown to activate genes involved in carbon metabolism and energy production during growth in glucose [10,31]. Previous work on Azf1p has been limited, so we began by monitoring the activity of *azf1Δ* in different carbon sources. A null mutant has previously been shown by spotting assay to produce a severe growth defect on non-fermentable carbon sources [31]. Slattery and colleagues also investigated the impact of the null mutant on growth on glucose by spotting assay, which did not reveal any differences between *azf1Δ* and the WT. We sought to further explore the role of Azf1p in glucose by monitoring growth in liquid media by optical density in a plate reader. We find that the null mutant has a shortened lag phase and reaches the log phase of growth more quickly when compared to the parent strain (Fig 1A). To further investigate the role of Azf1p during growth in glucose, we performed RNA-seq with RNA isolated from *azf1Δ* and WT cells grown in glucose and found many genes that are differentially expressed in the null mutant (Fig 1B, S1 Fig, S5 Table).

Genes downregulated by *azf1Δ* have not been previously reported, and we hypothesized that Azf1p may act as a co-repressor for these genes. The general co-repressor Cyc8p contains two Q-rich domains, one of which has been shown to mediate binding with cofactors [30]. Azf1p contains a polyQ and polyN domain (Fig 2A), and we hypothesized that one or both of these domains may be important for transcriptional repression by Azf1p. We first deleted the polyN domain using CRISPR (Fig 2A). We attempted to delete the polyQ domain using the same approach; however, deletion of the polyQ domain using this strategy was never successful. In an effort to probe the role of the polyQ domain, we deleted this domain from diploid BY4743. While the heterozygous mutant was viable, the strain did not sporulate. Null mutants are known to be sporulation deficient [31], and the sporulation deficiency observed in the heterozygous mutant suggests a fundamental role for the polyQ domain in Azf1p function.

### Differential roles of the polyN domain during growth in different carbon sources

PolyQ domains have been extensively studied due to their involvement in a number of mammalian neurodegenerative diseases. Despite being the second most common type of homorepeat in the eukaryotic proteome, the function of polyN repeats has been underexplored. We were therefore interested in characterizing the role of the polyN domain of Azf1p. The severe growth defect in glycerol media produced by the null mutant is one of Azf1p's most well-

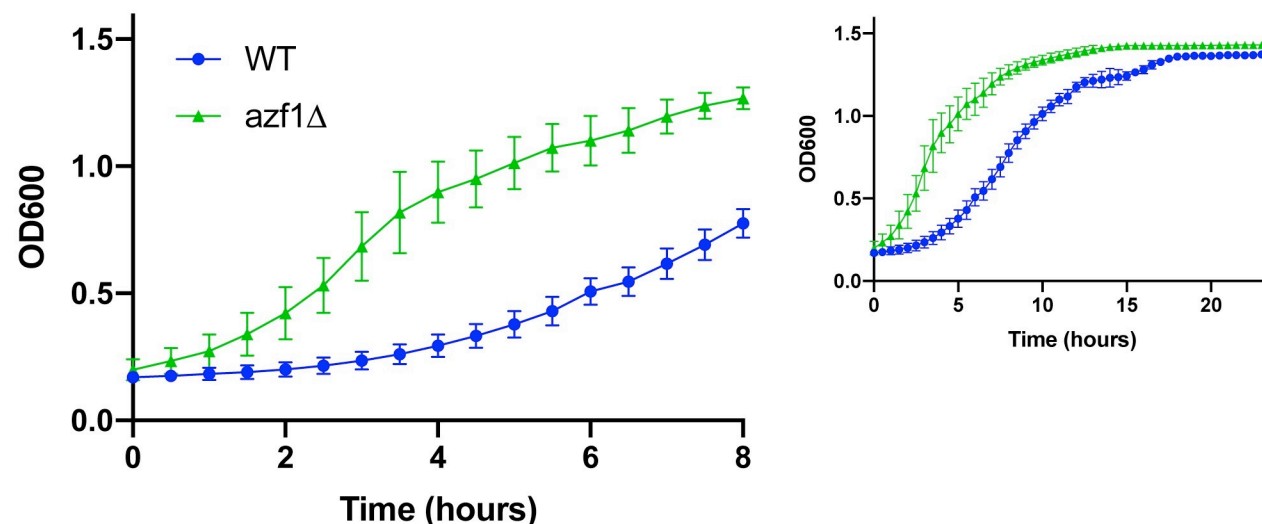

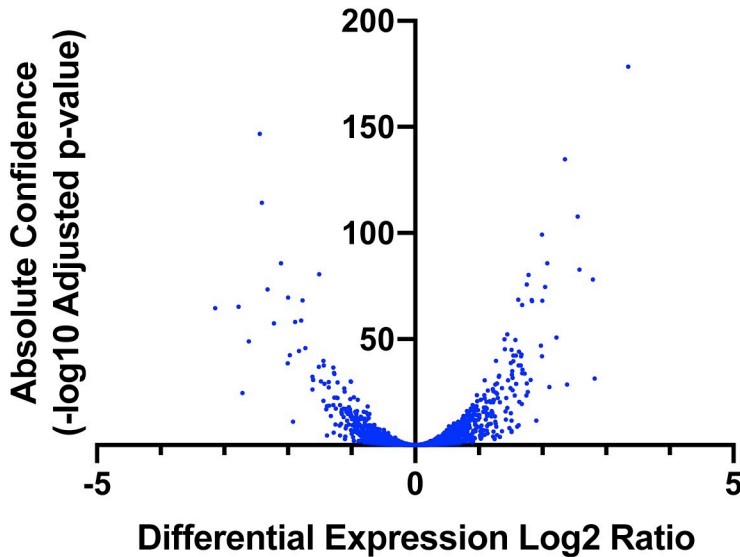

**Fig 1.** *azf1Δ* **shortens lag phase during growth in glucose.** A) Growth of WT and *azf1Δ* with glucose as the carbon source. Growth was measured by $OD_{600}$ in a plate reader for 24 hours. The first 8 hours are highlighted to show the differences in growth more clearly and due to the reduced accuracy of the plate reader at high optical densities. *azf1Δ* has a dramatically shorter lag phase than WT. Growth curves are the average of three biological replicates, and error bars indicate standard deviation. B) Volcano plot showing genes that are differentially expressed in *azf1Δ* compared to WT.

characterized phenotypes. We therefore measured the impact of deleting the polyN domain on growth in glycerol by spotting assay (Fig 2B). Deletion of the polyN domain produces a moderate growth defect on glycerol when compared to the parent strain. WT, ΔN, and *azf1Δ*

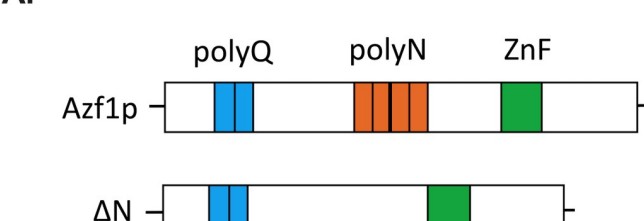

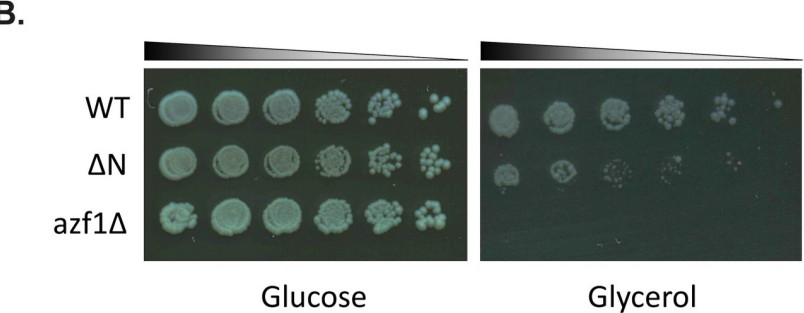

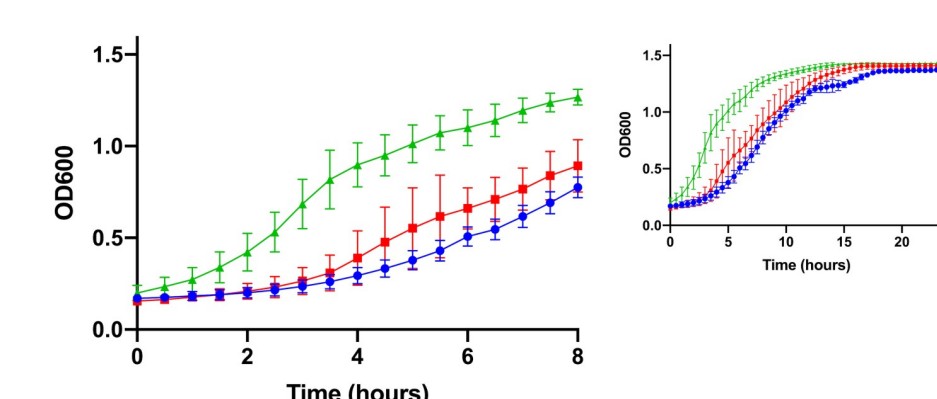

**Fig 2. The polyN domain of Azf1p impacts growth in different carbon sources.** A) Schematic of WT and ΔN Azf1p. Azf1p contains a polyQ (teal), polyN (orange), and a zinc-finger DNA-binding domain (ZnF) (green). To probe the role of the polyN domain, we deleted this domain in *AZF1-GFP* [32,33]. B) Spotting assay measuring the growth of WT, *ΔN*, and *azf1Δ* on glucose and glycerol. On glycerol, deletion of the polyN domain produces a mild growth defect, while *azf1Δ* produces a severe growth defect. Images are representative of 3 independent experiments. C) Growth of WT, ΔN, and *azf1Δ* with glucose as the carbon source. Growth was measured by OD$_{600}$ in a plate reader for 24 hours. The first 8 hours are highlighted to show the differences in growth more clearly and due to the reduced accuracy of the plate reader at high optical densities. Deletion of the polyN domain moderately shortens the lag phase, and *azf1Δ* dramatically shortens the lag phase compared to WT. Growth curves are the average of three biological replicates, and error bars indicate standard deviation.

appear to grow similarly on glucose, which is consistent with the finding by Slattery and colleagues that *AZF1* is not essential for growth in fermentable carbon sources. In order to detect any more subtle differences in growth between these strains in glucose, we monitored growth for 24 hours by optical density in a plate reader (Fig 1C). We find that deletion of the polyN

domain also reduces the length of the lag phase, although not to the same extent as the null mutant. This suggests that the polyN domain plays a fundamental role in Azf1p function, although deletion of this domain does not completely abolish Azf1p activity. To demonstrate that the growth phenotype observed in glucose is due to the deletion of the polyN domain, we repeated the plate reader assay with the WT and ΔN strains expressing a wildtype copy of *AZF1* from the pBY011-AZF1 plasmid. Expression of wildtype *AZF1* goes beyond rescuing the shortened lag phase phenotype and produces a growth defect in ΔN cells compared to WT (S3A Fig).

PolyN domains are prevalent in yeast TFs, but their function has only been explored in a few cases. In order to evaluate the role of the polyN domain in transcription, we performed RNA-seq using RNA isolated from WT and ΔN cells grown in both glucose and glycerol. We find that deletion of the polyN domain alters gene expression in both glucose (Fig 3A, S6 Table) and glycerol (Fig 3B, S7 Table) compared to WT. Notably, there are more statistically significantly differentially expressed genes in ΔN during growth in glucose compared to glycerol. The number of differentially expressed genes and the magnitudes of these changes, however, are much less than those observed in the null mutant, which indicates that the polyN domain plays a subtle Azf1p function.

In order to test the validity of the RNA-seq experiment, we measured relative gene expression of several of the most differentially expressed genes from each carbon source, as well as two known Azf1p target genes (*MDH2* and *GAS1*), by qPCR. Overall, the relative gene expression trends observed in ΔN and *azf1Δ* compared to WT are consistent with those measured by RNA-seq (S2 Fig).

In order to gain a better understanding of how the transcriptional changes in ΔN underlie the growth phenotypes observed in Fig 2B and 2C, we characterized differentially expressed genes with a log2 ratio of 0.30 or higher for upregulated genes and –0.30 or lower for downregulated genes by GO Enrichment Analysis [35–37]. Genes upregulated by ΔN during growth in glucose are enriched in two main biological processes: environmental stress tolerance and carbon metabolism and storage (Fig 3C). There are only 3 genes downregulated –0.30-fold or more during growth in glucose. As a result, there is no statistically significant biological process enrichment. Out of 119 genes upregulated 0.30-fold or higher in ΔN during growth in glycerol, 5 are involved in gluconeogenesis (17-fold enrichment), 8 are involved in glycolysis (15-fold enrichment), and 13 are involved in cell wall organization (3-fold enrichment). Out of 56 genes downregulated –0.30-fold or more in glycerol, 12 are involved in rRNA processing (4-fold enrichment) (Fig 3E).

## The polyN domain is dispensable for Azf1p localization

Azf1p has previously been shown to be localized to the nucleus during growth in glucose in both log and stationary phase, consistent with its function as a TF. Azf1p is also nuclear during log phase growth in glycerol, but relocalizes to cytoplasmic foci during stationary phase growth in glycerol [31]. Because polyN domains have previously been shown to promote protein aggregation [4,5], we hypothesized that the polyN domain is important for the formation of these foci. During growth in glucose, we find that both WT and ΔN localize to the nucleus (Fig 4A). In glycerol, both WT and ΔN localize to the nucleus during log phase, and both form cytoplasmic foci during stationary phase growth (Fig 4B). We therefore conclude that the polyN domain is not required for Azf1p relocalization during stationary phase growth in glycerol.

## [*AZF1*⁺] prion formation does not require the polyN domain

Azf1p was recently shown to form a prion, designated [*AZF1*⁺], that confers resistance to the drug radicicol. Both the polyQ and polyN domains are predicted to be capable of promoting

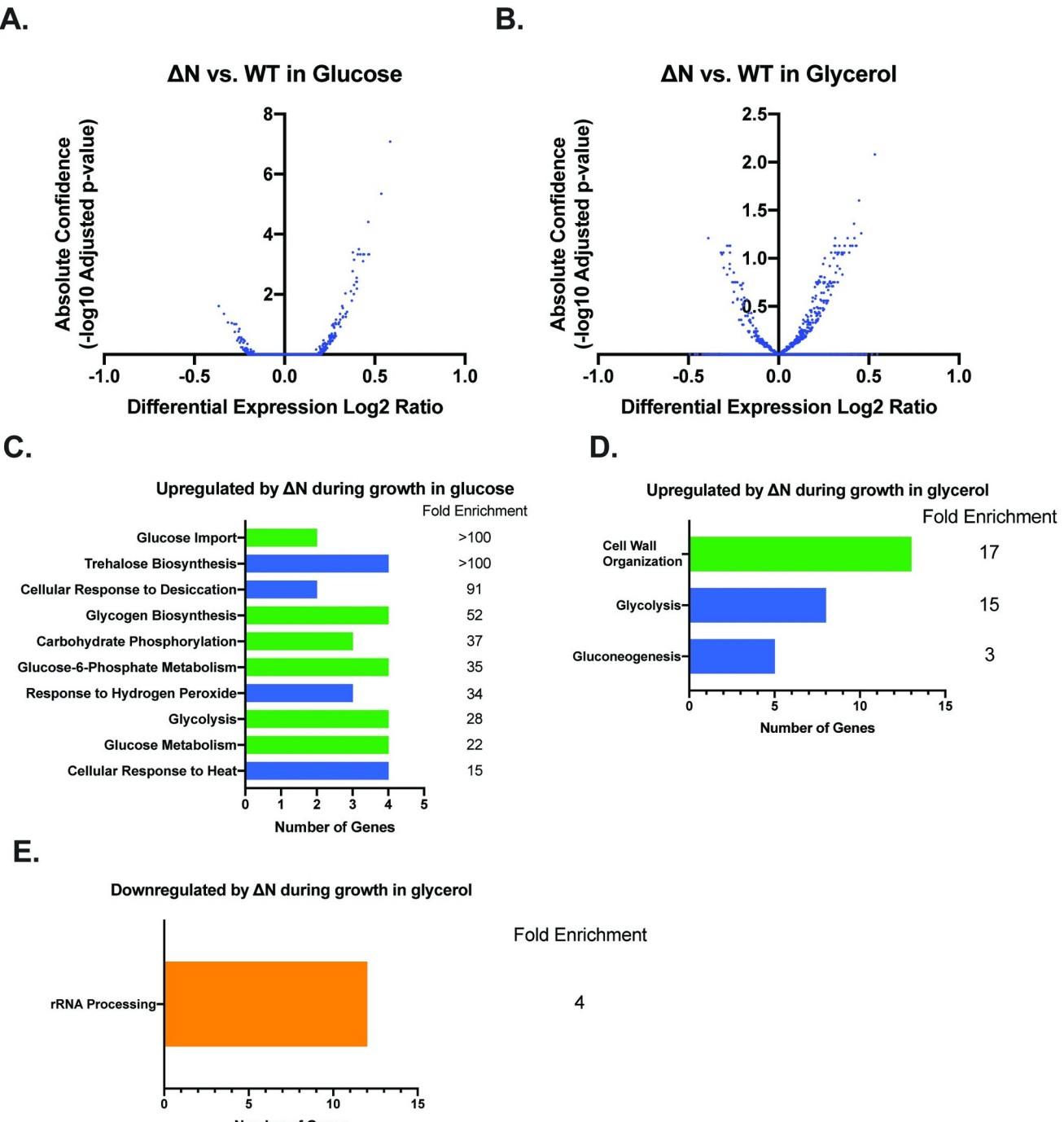

**Fig 3. The polyN domain of Azf1p plays a subtle role in transcription.** A) Volcano plot showing genes that are differentially expressed in ΔN compared to WT in glucose. B) Volcano plot showing genes that are differentially expressed in ΔN compared to WT in glycerol. C) GO Enrichment Analysis of upregulated genes with a log2 ratio of 0.30 or more in ΔN in glucose compared to WT. Fold enrichment is indicated to the right of each bar. Processes related to environmental stress tolerance are shown in blue, and processes related to carbon metabolism and storage are shown in green. D) GO Enrichment Analysis of upregulated genes with a log2 ratio 0.30 or higher in ΔN in glycerol compared to WT. Fold enrichment is indicated to the right of each bar. Processes related to environmental stress tolerance are shown in blue, and processes related to carbon metabolism and storage are shown in green. D) GO Enrichment Analysis of upregulated genes with a log2 ratio –0.30 or lower in ΔN in glycerol compared to WT. rRNA processing is the only process for which this set of genes is statistically significantly enriched (4-fold enrichment).

**A.**

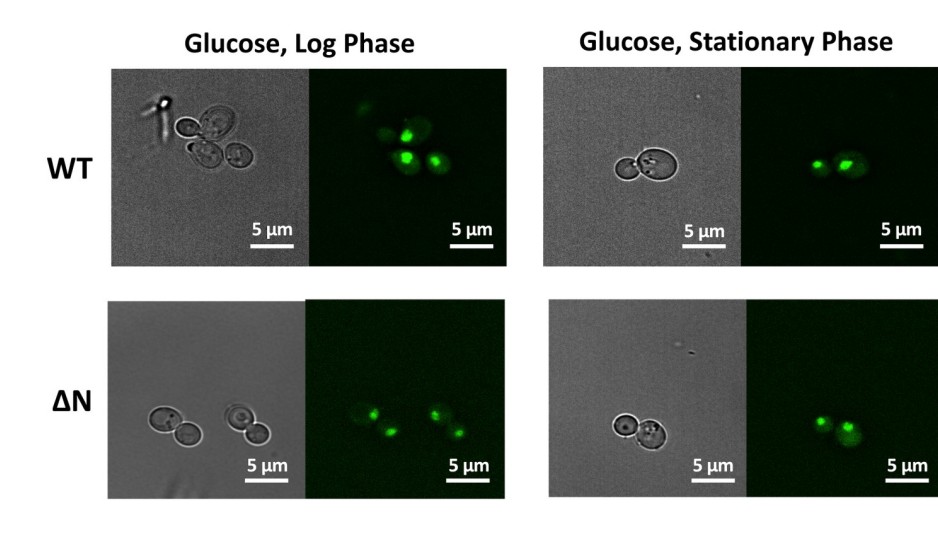

**B.**

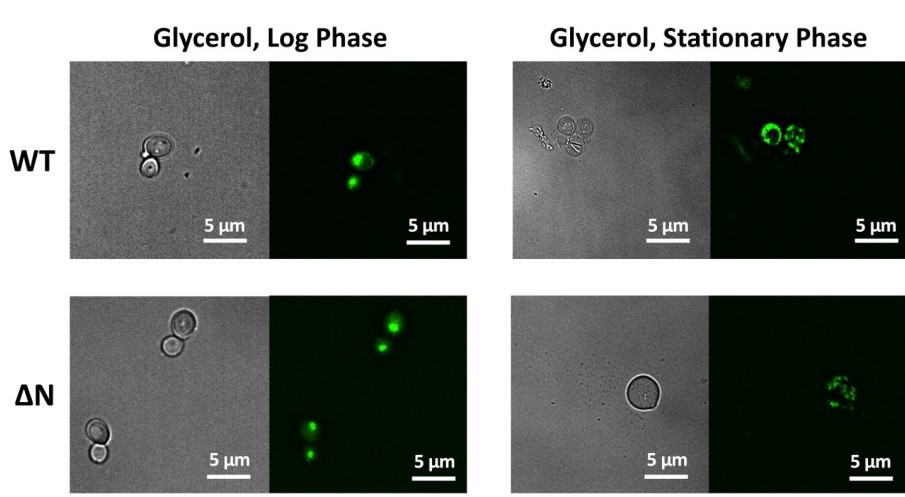

**Fig 4. Role of the polyN domain in localization of Azf1p.** A) Representative images of localization of Azf1p in glucose during growth in log and stationary phase. During growth in glucose, Azf1p is localized to the nucleus. B) Representative images of localization of Azfp1 during growth in glycerol in log and stationary phase. During log phase, Azf1p is localized to the nucleus. During stationary phase, both Azf1p and Azf1-ΔNp form cytoplasmic foci. At least 50 cells were imaged for each strain in each condition.

prion formation by the Prion-Like Amino Acid Composition (PLAAC) algorithm [39] (Fig 5A). Prion formation can be driven by overexpression of the constituent protein, and phenotypes conferred by the prion are stable over many generations after transient overexpression [10]. In order to measure the impact of deleting the polyN domain on prion formation, we measured resistance to radicicol of strains that experienced previous overexpression of *AZF1*. Consistent with previous reports, overexpression of WT improves growth in radicicol compared to cells that have never experienced overexpression (Fig 5B). Unexpectedly, overexpression of ΔN also confers increased resistance to radicicol compared to naïve cells (Fig 5C). Yeast prions have been referred to as a "bet-hedging" strategy due to their ability to confer beneficial phenotypes under some conditions and be deleterious in others [40]. We find that

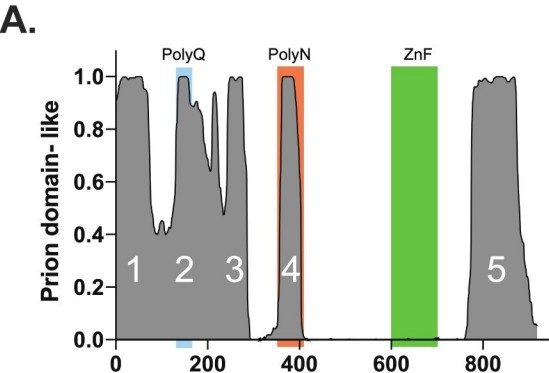

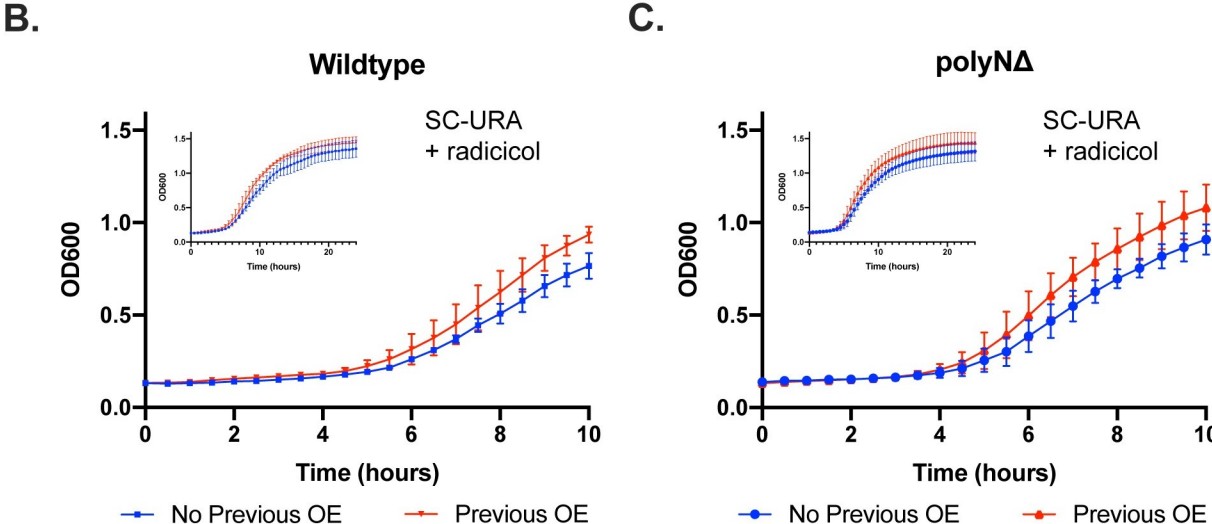

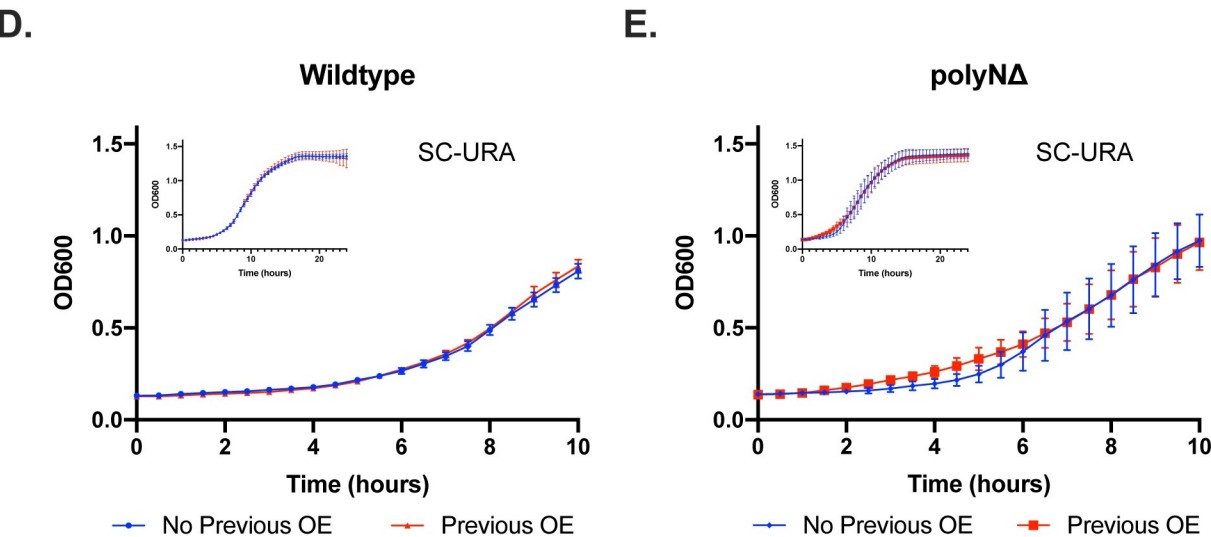

**Fig 5. Role of the polyN domain in prion formation and activity.** A) Azf1p domains predicted to be capable of promoting prion formation by PLAAC. The polyQ and polyN domains of Azf1p are both predicted to have prion-like properties. B) Growth of WT cells that did (red) and did

not (blue) experience previous overexpression (OE) of *AZF1* in SC-Ura + 75 μM radicicol measured by $OD_{600}$ in a plate reader. C) Growth of ΔN cells that did (red) and did not (blue) experience previous overexpression of *ΔN* in SC-Ura + 75 μM radicicol measured by $OD_{600}$ in a plate reader. Previous overexpression of both WT and ΔN confer resistance to radicicol. D) Growth of WT cells in SC-Ura with and without previous overexpression of *AZF1* measured by $OD_{600}$ in a plate reader. E) Growth ΔN cells in SC-Ura with and without previous overexpression of ΔN measured by $OD_{600}$ in a plate reader. Overexpression of *AZF1* does not impact growth under normal conditions. All growth curves are the average of three biological replicates. The first 10 hours are highlighted to show the differences in growth more clearly and due to the reduced accuracy of the plate reader at high optical densities.

transient overexpression of both WT and ΔN do not impact growth in the absence of radicicol (Fig 5D and 5E), indicating that [*AZF1*$^+$] is not toxic to cells under normal conditions.

## Azf1p exhibits repeat-length variation across strains of yeast

Repeat domains are well known to be genetically unstable, and changes in repeat copy number can impact protein function. This variation can enable organisms to adapt to different environmental conditions and drive evolution [30,41–46]. We therefore examined the homorepeat domains of Azf1p for length variation. Through a bioinformatic analysis of the genomes of 93 lab and wild isolates of *S. cerevisiae* [38,47], we determined that the polyN and polyQ domains of Azf1p vary in repeat copy number across strains of yeast. The polyN domain has 8 alleles ranging from 17 to 25 N residues (Fig 6A), and the polyQ domain has 3 alleles, 9Q, 12Q, and 17Q (Fig 6B).

Azf1p is a prion-forming yeast TF that contains a polyQ and a polyN domain. To begin to elucidate how repetitive regions modulate Azf1p function, in this work we sought to investigate the role of the polyN domain in transcription and [*AZF1*$^+$] prion formation. We find that the polyN domain plays a subtle role in transcription but is not required for prion formation. We also report that both the polyQ and polyN domains vary in copy number across strains of yeast.

## Discussion

Azf1p has previously been reported to be a transcriptional activator that is important for growth on glycerol and other non-fermentable carbon sources [31]. In this work, we report a new phenotype for *AZF1* during growth in glucose and identify a larger set of genes that are regulated by Azf1p. We monitored growth of WT and *azf1Δ* cells in glucose by optical density and find that the null mutant has a shortened lag phase compared to WT (Fig 1A). To further investigate the role of Azf1p during growth in glucose, we measured differential gene expression in the null mutant compared to WT by RNA-seq (Fig 1B and S5 Table). To our knowledge, *azf1Δ* has not been previously shown to upregulate gene expression. Our data suggests that Azf1p has a repressive function.

Despite being historically dismissed as functionally unimportant, homorepeats, particularly polyQ and polyN, are conserved across eukaryotic TFs and have been shown to play important roles in signaling, binding, and protein solubility. N-rich and Q-rich sequences are also characteristic of the prion-forming domains of yeast prions, and many repeat-containing TFs form prions. Azf1p contains a polyQ and a polyN repeat domain, and we sought to investigate the roles of these domains in transcription and prion activity. We attempted to delete each of these domains in the *AZF1-GFP* strain from the yeast GFP collection [32,33]. While deletion of the polyN domain was successful (Fig 2A), the polyQ domain could not be deleted using multiple strategies. This predicts a fundamental role for the polyQ domain of Azf1p in yeast but requires significant additional exploration that is beyond the scope of this manuscript. Here we continued the interrogation of the polyN domain as they are thought to serve similar functions to polyQ domains but are not nearly as well studied.

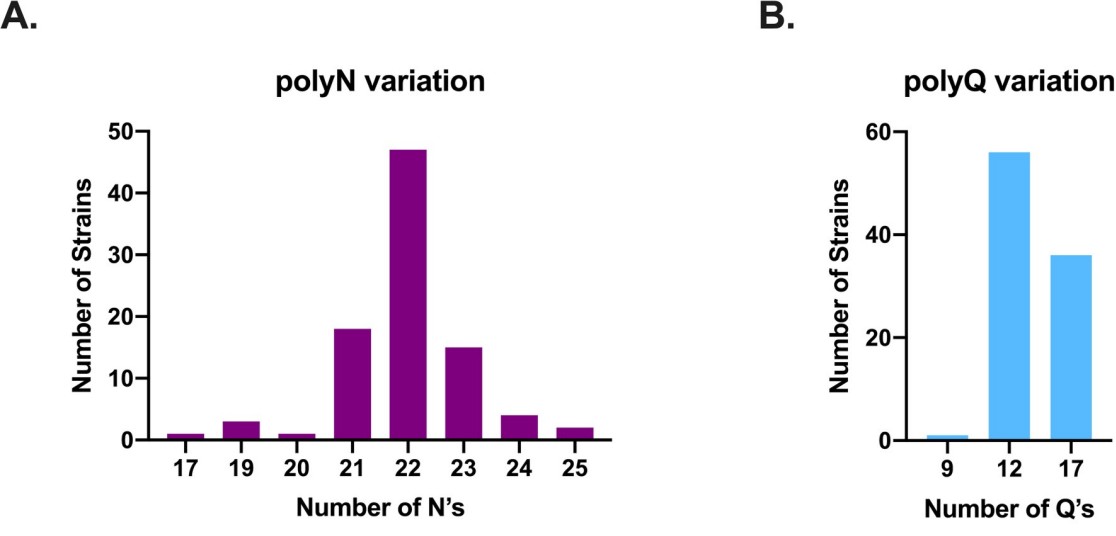

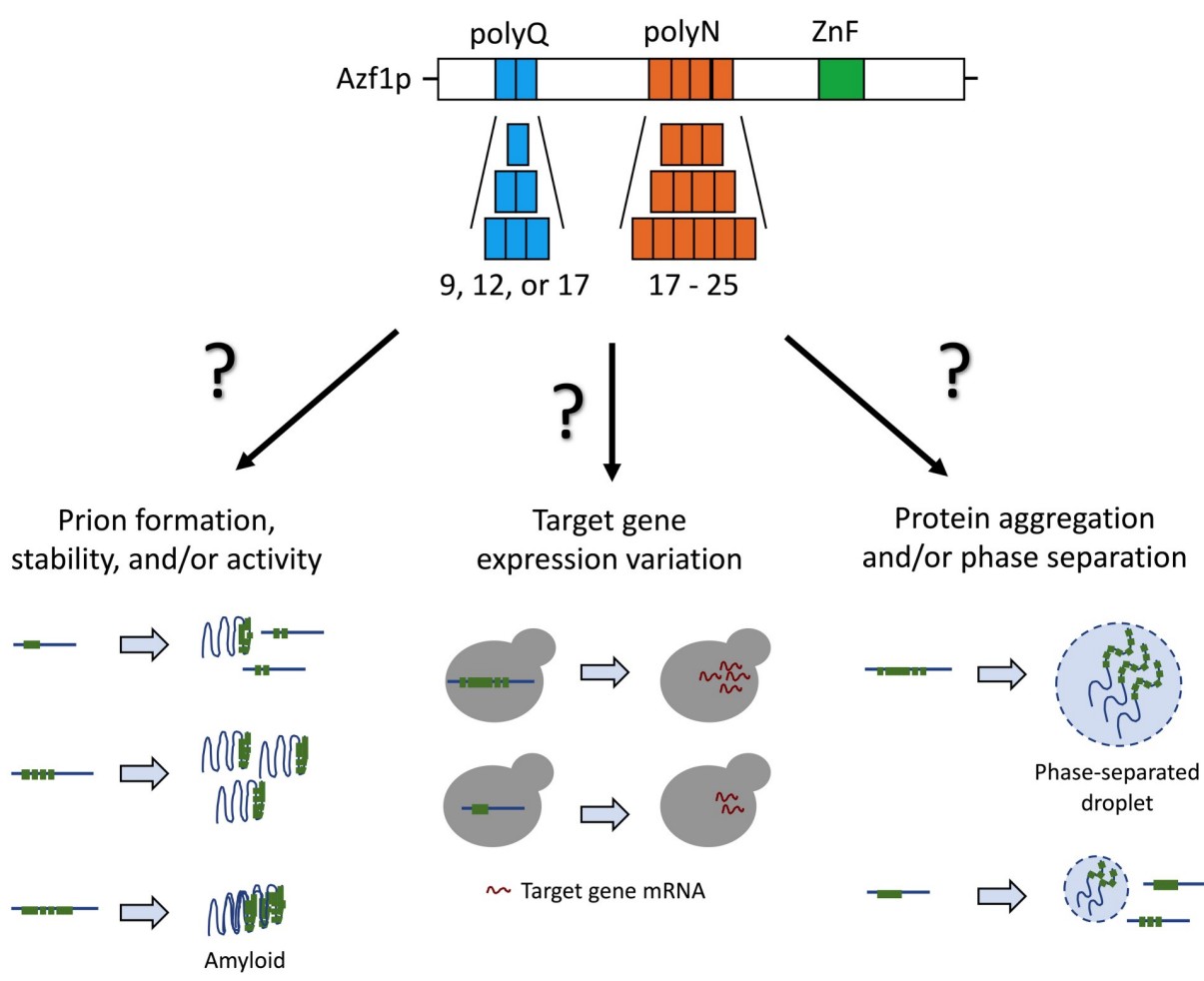

**Fig 6. Repeat variation in the polyQ and polyN domains of Azf1p may impact protein function.** A) The distribution of polyN domain variants across 93 strains of *S. cerevisiae*. The polyN domain has a range of alleles from 17 to 25 N residues. B) The distribution of polyQ domain variants across 93 strains of *S. cerevisiae*. The polyQ domain has 3 alleles: 9Q, 12Q, and 17Q. C) Model for ways that repeat copy number variation in Azf1p may impact protein function. Repeat variation may impact prion formation and/or stability, target gene expression, or protein aggregation and/or phase separation.

Due to the known involvement of a homorepeat domain in the repressive activity of Cyc8p, we hypothesized that the polyN domain may be important for this function. In order to address this hypothesis, we first examined the role of the polyN domain in growth in different carbon sources. We find that deletion of the polyN domain produces a mild growth defect on glycerol and moderately reduces the length of lag phase during growth in glucose compared to WT (Fig 2B and 2C). Because each of these phenotypes is less pronounced than the corresponding phenotype observed in the null mutant, we conclude that the polyN domain has a minor but not essential role in Azf1p function. In order to verify that the shortened lag phase observed in glucose is due to the deletion of the polyN domain, we attempted to rescue this phenotype by expressing a wildtype copy of *AZF1* from the pBY011-AZF1 plasmid. While this plasmid encodes *AZF1* under the control of a galactose-inducible promoter, we have determined by qPCR that *AZF1* is expressed from this plasmid at a low level in glucose. In the WT strain, we compared the relative expression of *AZF1* in cells transformed with pBY011-AZF1 to cells transformed with pBY011-GFP, which encodes GFP under the galactose-inducible promoter instead of *AZF1*. We find that during growth in glucose, pBY011-AZF1 results in significantly increased expression of *AZF1* (S3B Fig). Even with this low level of leaky expression, we find that the plasmid copy of wildtype *AZF1* goes beyond rescuing the shortened lag phase phenotype and produces a growth defect in ΔN cells compared to WT (S3A Fig). While we recognize that rescuing the observed ΔN growth phenotypes with a wildtype copy of *AZF1* under the native promoter would be more appropriate, the constraints of the COVID-19 pandemic prevented us from performing this experiment.

In order to better understand the observed growth phenotypes, we compared gene expression levels in ΔN to WT as measured by RNA-seq. Consistent with our phenotypic growth data, we find that deletion of the polyN domain impacts gene expression in both glucose (Fig 3A) and glycerol (Fig 3B), suggesting that this domain is involved in Azf1p's TF activity. During growth in both carbon sources, more genes are upregulated than downregulated upon deletion of the polyN domain. This result is consistent with our hypothesis that the polyN domain plays a role in Azf1p's repressive function.

In order to explore how the changes in gene expression measured by RNA-seq give rise to the observed growth patterns in each carbon source, we performed GO Enrichment Analysis on genes differentially expressed between ΔN and WT. In glycerol, genes upregulated by ΔN are enriched in gluconeogenesis (17-fold enrichment), glycolysis (15-fold enrichment), and cell wall organization (3-fold enrichment) (Fig 3D). These processes are consistent with the those previously reported to be regulated by Azf1p [31]. Genes upregulated by ΔN during growth in glucose fit into two main categories: environmental stress tolerance and carbon metabolism and storage (Fig 3C). Upregulation of carbon metabolism genes may underlie the shortened lag phase observed in ΔN, as cells that can import and utilize glucose more rapidly or efficiently may be able to enter the log phase of growth more quickly. This result may have implications for fermentation applications and small molecule production. Yeast are used in the production of a variety of foods and drinks, as well as natural and non-natural small molecules for pharmaceuticals, fuels, and even perfumes [48–50]. Metabolic engineering is a crucial technology geared towards improving the efficiency and output of these processes, and the new phenotypes we report for a null mutant and a ΔN mutant indicate that Azf1p may be a useful target.

We hypothesized that the transcriptional changes observed in ΔN may result from altered Azf1p localization. During both log and stationary phase growth, we find that ΔN exhibits the same localization patterns as WT regardless of the carbon source (Fig 4). Deletion of the polyN domain therefore does not appear to impact Azf1p localization. This observation is consistent with the result that deletion of the polyN domain impacts the expression of only a subset of Azf1p's target genes, as altered localization would likely affect all of Azf1p's targets. The polyN domain may instead act as a binding domain with transcriptional cofactors. This hypothesis is supported by the different growth and gene expression patterns observed between ΔN and *azf1Δ*. Different target genes have different combinations of transcriptional regulators. The polyN domain may be important for binding with only a subset of Azf1p's cofactors, causing deletion of this domain to impact expression of only a subset of Azf1p's target genes.

In a large-scale screen for yeast prions conducted by Chakrabortee and colleagues, Azf1p was found to form a prion, [*AZF1*+], that confers resistance to the drug radicicol. Overexpression of the constituent protein drives prion formation, and phenotypes conferred by the prion are stable over many generations after transient overexpression [10]. The PrD of Azf1p has not been characterized experimentally, but PLAAC predicts that the polyN domain is one of several sequences capable of promoting prion formation [39] (Fig 5A). In order to determine if the polyN domain plays a role in [*AZF1*+] prion formation or activity, we monitored growth in radicicol of WT and ΔN cells that previously overexpressed WT and ΔN, respectively. Consistent with Chakrabortee and colleagues' results, transient overexpression of WT improves growth in radicicol compared to cells that have never experienced overexpression (Fig 5B). Surprisingly, overexpression of ΔN also confers resistance to radicicol compared to naïve cells (Fig 5C), suggesting that the polyN domain is dispensable for prion formation. The polyQ domain is within a large N-terminal IDR predicted by PLAAC to be PrD-like [39] (Fig 5A). This domain may therefore be a more relevant candidate for future studies aimed at determining the PrD of Azf1p.

Finally, we report that the polyN and polyQ domains of Azf1p vary in repeat copy number across strains of yeast. The polyN domain has 8 alleles ranging from 17 to 25 N residues (Fig 6A), and the polyQ domain has 3 alleles, 9Q, 12Q, and 17Q (Fig 6B). We propose a model outlining several ways that repeat-length variation may impact Azf1p function (Fig 6C). First, repeat variation could alter prion formation and/or stability. In yeast, expanded repeats in PrDs have been shown to increase prion formation [51–53]. While we find that the polyN domain does not play a significant role in prion formation, PLAAC predicts that the polyQ domain is PrD-like (Fig 5A). If the polyQ domain is in fact important for prionogenesis, short polyQ alleles may reduce prion formation and/or stability of the prion conformer, while long alleles or abnormally expanded repeats could result in the formation of an irreversible amyloid fiber.

Repeat variation could also impact expression of Azf1p's target genes. Another prion-forming yeast TF, Cyc8p, has a naturally variable polyQ domain. Variation in this domain impacts Cyc8p solubility and interaction with its binding partners, which in turn alters target gene expression and the associated phenotypes [30]. We find that the polyN domain of Azf1p plays a subtle role in transcription, possibly by mediating binding to transcriptional cofactors. The impact of repeat variation in the polyN domain of Azf1p, as well as the variable repeat domains of many other TFs across eukaryotes, on target gene expression warrants further study. The gene expression and resulting metabolic changes we observed in ΔN may have implications in fermentation applications and small molecule production. Deletion of the polyN domain moderately shortens the lag phase in glucose (Fig 1C), a phenotype that can potentially be explained by the upregulation of genes involved in carbon metabolism and storage (Fig 3C). Repeat variation may serve as a mechanism to further tune this phenotype by altering gene expression.

One possible mechanism by which repeat variation could impact Azf1p's function in transcription is phase separation. Several eukaryotic TFs have been shown to phase separate with Mediator via their IDRs, and the formation of these droplets was demonstrated to be required for transcription [54,55]. The polyQ and polyN domains of Azf1p are both intrinsically disordered [47], and length variation within these domains could impact the extent to which Azf1p can be incorporated into phase separated droplets.

Repetitive domains were historically dismissed as functionally unimportant due to their lack of three-dimensional structure. These domains are now known to be important for signaling, binding, prion formation, and phase separation. Homorepeats are particularly common in eukaryotic TFs and often exhibit length variation across organisms. The function of these domains and the impact of repeat variation, however, has only been minimally explored. We report a new role for the variable polyN domain of the yeast TF Azf1p in transcription and highlight repeat copy number variation as a possible driver of target gene expression variance in this and other eukaryotic TFs. We also report metabolic changes that may be useful for fermentation applications and small molecule production.

## Supporting information

**S1 Fig. GO Enrichment Analysis of genes differentially expressed in *azf1Δ* during growth in glucose.** A) GO Enrichment Analysis of genes that are statistically significantly upregulated by *azf1Δ* (p<0.001) compared to WT as measured by RNA-seq with RNA isolated from cells grown in glucose. B) GO Enrichment Analysis of genes that are statistically significantly downregulated by *azf1Δ* compared to WT (p<0.001) as measured by RNA-seq using RNA isolated from cells grown in glucose.
(TIF)

**S2 Fig. Relative gene expression analysis of Azf1p target genes by qPCR.** A) Fold change in expression of Azf1p target genes in *AZF1ΔN* and *azf1Δ* compared to WT during growth in glucose. B) Fold change in expression of Azf1p target genes in *AZF1ΔN* compared to WT during growth in glycerol. Error bars represent standard deviation. Statistical significance was calculated using an unpaired student's t test. * indicates p<0.05 and ** indicates p<0.001.
(TIF)

**S3 Fig. Expression of wildtype *AZF1* abolishes the shortened lag phase observed in ΔN cells during growth in glucose.** A) Growth of WT (blue) and ΔN (red) cells with glucose as the carbon source. Growth was measured by $OD_{600}$ in a plate reader for 24 hours. Growth curves are the average of three biological replicates, and error bars indicate standard deviation. B) Fold change in expression of *AZF1* in WT cells expressing pBY011-AZF1 compared to pBY011-GFP during growth in glucose. Error bars represent standard deviation. Statistical significance was calculated using an unpaired student's t test (p = 0.0028).
(TIF)

**S1 Table. Plasmids used in this work.**
(PDF)

**S2 Table. Primers used in this work.**
(PDF)

**S3 Table. Yeast strains used in this work.**
(PDF)

**S4 Table. Fold change in *AZF1* expression in 3 biological replicates of *azf1Δ* compared to WT.**
(PDF)

**S5 Table. Differentially expressed genes in *azf1Δ* compared to WT during growth in glucose.**
(CSV)

**S6 Table. Differentially expressed genes in ΔN compared to WT during growth in glucose.**
(CSV)

**S7 Table. Differentially expressed genes in ΔN compared to WT during growth in glycerol.**
(CSV)

# Acknowledgments

The authors would like to acknowledge Mackenzie Parmenter and Mike Babokhov for early contributions to this project. We thank M. McVey, K.A. McElroy, and members of the McVey Lab and Tufts Biology department for their helpful discussions during the preparation of this manuscript.

# Author Contributions

**Conceptualization:** Taylor Stewart, Stephen M. Fuchs.

**Data curation:** Taylor Stewart, Benjamin E. Wolfe.

**Funding acquisition:** Stephen M. Fuchs.

**Investigation:** Taylor Stewart, Stephen M. Fuchs.

**Methodology:** Taylor Stewart, Benjamin E. Wolfe, Stephen M. Fuchs.

**Project administration:** Stephen M. Fuchs.

**Resources:** Stephen M. Fuchs.

**Supervision:** Stephen M. Fuchs.

**Writing – original draft:** Taylor Stewart, Stephen M. Fuchs.

**Writing – review & editing:** Taylor Stewart, Benjamin E. Wolfe, Stephen M. Fuchs.

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
