## [Decision Letter · Decision Letter 0]

22 Feb 2021

PONE-D-21-03428

Defining the role of the polyasparagine repeat domain of the S. cerevisiae transcription factor Azf1p

PLOS ONE

Dear Dr. Fuchs,

Thank you for submitting your manuscript to PLOS ONE. After careful consideration, we feel that it has merit but does not fully meet PLOS ONE’s publication criteria as it currently stands. Therefore, we invite you to submit a revised version of the manuscript that addresses the points raised during the review process.

We look forward to receiving your revised manuscript.

Kind regards,

Alvaro Galli

Academic Editor

PLOS ONE

Additional Editor Comments:

Journal Requirements:

"This work was supported by the Army Research Office [W911NF-16-1-0175, W911NF-19-1-

0299, and W911NF-20-1-0083 to S.M.F.]. The funders had no role in study design, data

collection and"

Reviewers' comments:

Reviewer's Responses to Questions

**Comments to the Author**

1. Is the manuscript technically sound, and do the data support the conclusions?

Reviewer #1: Yes

Reviewer #2: Yes

2. Has the statistical analysis been performed appropriately and rigorously? 

Reviewer #1: Yes

Reviewer #2: Yes

3. Have the authors made all data underlying the findings in their manuscript fully available?

Reviewer #1: Yes

Reviewer #2: Yes

4. Is the manuscript presented in an intelligible fashion and written in standard English?

Reviewer #1: Yes

Reviewer #2: Yes

5. Review Comments to the Author

Reviewer #1: This manuscript presents an analysis of the role of the poly-asparagine (polyN) domain of the yeast protein Azf1p. The authors report a role of the polyN domain in mediating Azf1p's function as a transcriptional regulator of carbon metabolism. They show that the polyN domain is not required for prion-like formation in yeast and provide a bioinformatics analysis across diverse yeast strains revealing diversity in length of the repeat across different yeast strains. Overall the experiments are performed to a high standard and are rigorous and the conclusions are supported by the data. I therefore recommend publication of this study in PLoS One because it fulfills the requirements for publication in this journal. I have one suggestion for the authors to consider. They may want to to perform rescue experiments with WT Azf1p (e.g., from a plasmid) to make sure the growth phenotypes and the changes in gene expression that they report in their ∆N mutants are definitely caused by the mutation and not some other perturbation.

Reviewer #2: Dear Editor

In the present paper by Stewart et al, the authors analyze the function of azf1 protein, a Zinc-finger transcription factor, regulating the expression of genes involved in growth, carbon metabolism and in cell wall integrity depending on carbon source present in growth medium. The protein contains polyN and polyQ domain typical of prions. Indeed it has been shown that azf1 exists in prion state and confer resistance to the drug radicicol. The aim of the paper is to characterize the polyN domain of azf1 and define its role in transcription and prion formation. The authors demonstrate that this domain has a subtle role in transcription but not in prion formation. Investigations on yeast prion has an important spill-over on the improvement of yeast strains for fermentation applications and small molecule production.

The paper is well written and clear and experiments are carefully designed and mapped out. Moreover Data are convincing, well presented and discussed.

I have only a minor drawbacks that need to be addressed before publication.

1- Pag 4 at the end of introduction “cell well integrity” must be corrected in “cell wall integrity”

2- Pag 10 paragraph “Differential roles of the polyN domain during growth in different carbon sources”:

The concept expressed in line 3-4 referring to the phenotype of azf1delta strain is then repeated in line 6 with the sentence “ Consistent with previous reports, azf1∆ does not support growth on glycerol”. I think that the latter sentence can be omitted.

6. PLOS authors have the option to publish the peer review history of their article (what does this mean?). If published, this will include your full peer review and any attached files.

Reviewer #1: **Yes: **Aaron Gitler (Stanford University)

Reviewer #2: No

---

## [Author Response · Author response to Decision Letter 0]

8 Apr 2021

Response to Reviewers:

Reviewer #1: 

This manuscript presents an analysis of the role of the poly-asparagine (polyN) domain of the yeast protein Azf1p. The authors report a role of the polyN domain in mediating Azf1p's function as a transcriptional regulator of carbon metabolism. They show that the polyN domain is not required for prion-like formation in yeast and provide a bioinformatics analysis across diverse yeast strains revealing diversity in length of the repeat across different yeast strains. 

Overall the experiments are performed to a high standard and are rigorous and the conclusions are supported by the data. I therefore recommend publication of this study in PLoSOne because it fulfills the requirements for publication in this journal. 

Many thanks

I have one suggestion for the authors to consider. They may want to to perform rescue experiments with WT Azf1p (e.g., from a plasmid) to make sure the growth phenotypes and the changes in gene expression that they report in their ∆N mutants are definitely caused by the mutation and not some other perturbation.

This was a fantastic suggestion by Reviewer 1 and one which we spent the past several weeks trying to implement. We have included now an additional figure (S3 Fig) and updated the text to show that indeed the growth phenotypes are caused by the ∆N mutations. 

Reviewer #2: 

In the present paper by Stewart et al, the authors analyze the function of azf1 protein, a Zinc-finger transcription factor, regulating the expression of genes involved in growth, carbon metabolism and in cell wall integrity depending on carbon source present in growth medium. The protein contains polyN and polyQ domain typical of prions. Indeed it has been shown that azf1 exists in prion state and confer resistance to the drug radicicol. The aim of the paper is to characterize the polyN domain of azf1 and define its role in transcription and prion formation. The authors demonstrate that this domain has a subtle role in transcription but not in prion formation. Investigations on yeast prion has an important spill-over on the improvement of yeast strains for fermentation applications and small molecule production.

The paper is well written and clear and experiments are carefully designed and mapped out. Moreover Data are convincing, well presented and discussed.

Much appreciated 

I have only a minor drawbacks that need to be addressed before publication.

1- Pag 4 at the end of introduction “cell well integrity” must be corrected in “cell wall integrity”

Fixed

2- Pag 10 paragraph “Differential roles of the polyN domain during growth in different carbon sources”:

The concept expressed in line 3-4 referring to the phenotype of azf1delta strain is then repeated in line 6 with the sentence “ Consistent with previous reports, azf1∆ does not support growth on glycerol”. I think that the latter sentence can be omitted.

Also corrected in the text

---

## [Decision Letter · Decision Letter 1]

13 Apr 2021

PONE-D-21-03428R1

Defining the role of the polyasparagine repeat domain of the S. cerevisiae transcription factor Azf1p

PLOS ONE

Dear Dr. Fuchs,

Thank you for submitting your manuscript to PLOS ONE. After careful consideration, we feel that it has merit but does not fully meet PLOS ONE’s publication criteria as it currently stands. Therefore, we invite you to submit a revised version of the manuscript that addresses the points raised during the review process.

Please note that reviewer 2 has still some minor concerns that need to be addressed to accept the manuscript

We look forward to receiving your revised manuscript.

Kind regards,

Alvaro Galli

Academic Editor

PLOS ONE

Journal Requirements:

Reviewers' comments:

Reviewer's Responses to Questions

**Comments to the Author**

1. If the authors have adequately addressed your comments raised in a previous round of review and you feel that this manuscript is now acceptable for publication, you may indicate that here to bypass the “Comments to the Author” section, enter your conflict of interest statement in the “Confidential to Editor” section, and submit your "Accept" recommendation.

Reviewer #1: All comments have been addressed

Reviewer #2: All comments have been addressed

2. Is the manuscript technically sound, and do the data support the conclusions?

Reviewer #1: Yes

Reviewer #2: Yes

3. Has the statistical analysis been performed appropriately and rigorously? 

Reviewer #1: Yes

Reviewer #2: N/A

4. Have the authors made all data underlying the findings in their manuscript fully available?

Reviewer #1: Yes

Reviewer #2: Yes

5. Is the manuscript presented in an intelligible fashion and written in standard English?

Reviewer #1: Yes

Reviewer #2: Yes

6. Review Comments to the Author

Reviewer #1: (No Response)

Reviewer #2: The authors addressed all comments, but I still have two minor concerns:

-I could not see the new figure S3 because, as the other supplementary figures, was not loaded

-The authors expressed AZF1 from a plasmid where the gene is under control of GAL promoter. However, by qPCR, they observed expression of the gene also in glucose and this determined the rescue of the phenotype. It would be interesting to know the expression level of azf1 in glucose so I would add in the figure s3 the results of qPCR because, although it is known that gal promoter is leaky, the use of it to express protein in medium with glucose is not common.

7. PLOS authors have the option to publish the peer review history of their article (what does this mean?). If published, this will include your full peer review and any attached files.

Reviewer #1: **Yes: **Aaron Gitler

Reviewer #2: No

---

## [Author Response · Author response to Decision Letter 1]

19 Apr 2021

Reviewer #1: (No Response)

Reviewer #2: The authors addressed all comments, but I still have two minor concerns:

-I could not see the new figure S3 because, as the other supplementary figures, was not loaded

We have uploaded new versions of the supplemental figures (which include the additional data asked for below). The manuscript should now be complete.

-The authors expressed AZF1 from a plasmid where the gene is under control of GAL promoter. However, by qPCR, they observed expression of the gene also in glucose and this determined the rescue of the phenotype. It would be interesting to know the expression level of azf1 in glucose so I would add in the figure s3 the results of qPCR because, although it is known that gal promoter is leaky, the use of it to express protein in medium with glucose is not common.

We have added the qPCR data as reviewer 2 suggested to Figure S3B. We agree, this is not common, but the best we were able to use under our current working circumstances.

---

## [Decision Letter · Decision Letter 2]

26 Apr 2021

Defining the role of the polyasparagine repeat domain of the S. cerevisiae transcription factor Azf1p

PONE-D-21-03428R2

Dear Dr. Fuchs,

We’re pleased to inform you that your manuscript has been judged scientifically suitable for publication and will be formally accepted for publication once it meets all outstanding technical requirements.

Kind regards,

Alvaro Galli

Academic Editor

PLOS ONE

Additional Editor Comments (optional):

Reviewers' comments:

Reviewer's Responses to Questions

**Comments to the Author**

1. If the authors have adequately addressed your comments raised in a previous round of review and you feel that this manuscript is now acceptable for publication, you may indicate that here to bypass the “Comments to the Author” section, enter your conflict of interest statement in the “Confidential to Editor” section, and submit your "Accept" recommendation.

Reviewer #1: All comments have been addressed

Reviewer #2: All comments have been addressed

2. Is the manuscript technically sound, and do the data support the conclusions?

Reviewer #1: Yes

Reviewer #2: Yes

3. Has the statistical analysis been performed appropriately and rigorously? 

Reviewer #1: Yes

Reviewer #2: Yes

4. Have the authors made all data underlying the findings in their manuscript fully available?

Reviewer #1: Yes

Reviewer #2: Yes

5. Is the manuscript presented in an intelligible fashion and written in standard English?

Reviewer #1: Yes

Reviewer #2: Yes

6. Review Comments to the Author

Reviewer #1: (No Response)

Reviewer #2: (No Response)

7. PLOS authors have the option to publish the peer review history of their article (what does this mean?). If published, this will include your full peer review and any attached files.

Reviewer #1: **Yes: **Aaron Gitler

Reviewer #2: No

---

## [Editor Report · Acceptance letter]

14 May 2021

PONE-D-21-03428R2 

Defining the role of the polyasparagine repeat domain of the *S. cerevisiae* transcription factor Azf1p 

Dear Dr. Fuchs:

I'm pleased to inform you that your manuscript has been deemed suitable for publication in PLOS ONE. Congratulations! Your manuscript is now with our production department. 

Kind regards, 

on behalf of

Dr. Alvaro Galli 

Academic Editor

PLOS ONE